# Phobalysin: Fisheye View of Membrane Perforation, Repair, Chemotaxis and Adhesion

**DOI:** 10.3390/toxins11070412

**Published:** 2019-07-16

**Authors:** Gisela von Hoven, Amable J. Rivas, Matthias Husmann

**Affiliations:** Institute of Medical Microbiology and Hygiene, University Medical Center, Johannes Gutenberg-University, 55128 Mainz, Germany

**Keywords:** *Photobacterium damselae *subsp. *damselae*, phobalysin, pore forming toxin, membrane repair, adhesion, chemotaxis

## Abstract

Phobalysin P (PhlyP, for photobacterial lysin encoded on a plasmid) is a recently described small β-pore forming toxin of *Photobacterium damselae* subsp. *damselae (Pdd).* This organism, belonging to the family of *Vibrionaceae,* is an emerging pathogen of fish and various marine animals, which occasionally causes life-threatening soft tissue infections and septicemia in humans. By using genetically modified *Pdd* strains, PhlyP was found to be an important virulence factor. More recently, in vitro studies with purified PhlyP elucidated some basic consequences of pore formation. Being the first bacterial small β-pore forming toxin shown to trigger calcium-influx dependent membrane repair, PhlyP has advanced to a revealing model toxin to study this important cellular function. Further, results from co-culture experiments employing various *Pdd* strains and epithelial cells together with data on other bacterial toxins indicate that limited membrane damage may generally enhance the association of bacteria with target cells. Thereby, remodeling of plasma membrane and cytoskeleton during membrane repair could be involved. In addition, a chemotaxis-dependent *attack-and track* mechanism influenced by environmental factors like salinity may contribute to PhlyP-dependent association of *Pdd* with cells. Obviously, a synoptic approach is required to capture the regulatory links governing the interaction of *Pdd* with target cells. The characterization of *Pdd’s* secretome may hold additional clues because it may lead to the identification of proteases activating PhlyP’s pro-form. Current findings on PhlyP support the notion that pore forming toxins are not just killer proteins but serve bacteria to fulfill more subtle functions, like accessing their host.

## 1. Introduction

*Photobacterium damselae* subsp. *damselae (Pdd*) belongs to the family of *Vibrionaceae*, aquatic bacteria widespread in coastal and estuarine environments which cause a variety of diseases in vertebrates and invertebrates, including gastrointestinal disease, skin ulceration, soft tissue infections and septicemia [1]. *Pdd* is a halophilic bacterium that infects fish, and a range of other marine animals. From time to time serious soft tissue infections and septicemia by *Pdd* in humans are being reported (e.g., [2,3,4,5]). The generalist character of the organism, its impressive armory, and taxonomical and epidemiological aspects have been discussed in previous reviews [6,7,8]. Below, we will introduce the organism and its role as an infectious agent, with emphasis on human disease. Next, *Pdd´*s hemolysins and genetic approaches to investigate these toxins will be discussed to then focus on phobalysin P (PhlyP), a recently identified pore forming toxin (PFT) of *Pdd* which is a virulence factor and a promising tool in cell biology.

*Pdd,* originally termed *Vibrio damsela,* was discovered in 1981 as a pathogen of the blacksmith, a temperate-water damselfish (*Chromis punctipinnis*), which lives along inshore reefs of the Californian coast between Monterey and Baja California [9]. The pioneering study was undertaken to clarify the etiology of ulcers that occur along the flank of blacksmith captured during the spawning season in summer and fall (such ulcers are unusual among marine fish). Cultures of swabs from ulcers yielded a variety of bacterial species and protozoa. The bacterial isolates were tested for their ability to produce ulcers upon infection of scarified dermis of healthy *C. punctipinnis*. Among them, only *V. damsela*, a new species, proved to be pathogenic, and infection with 10^7^–10^8^ colony forming units (CFU) bacteria produced large ulcers within 3 days. Further experiments revealed that it was not even necessary to scarify the dermis of fish to cause infections [9]. Initially, *V. damsela* was not detected in several other fish species inhabiting the same reefs populated by *C. punctipinnis*, but in subsequent studies the organism was isolated from a multitude of marine animals. Thus, *V. damsela* was found in Hawaiian green turtles [10], shrimp [11], food fish [12,13], prawns, and neritic sharks [14]. *Pdd* infections have been reported from the western and eastern hemisphere–from northerly located countries like Denmark [12] as well as from southern places like Tasmania [15]. Infections occurred in salt water of coastal areas and marine farms, in brackish water of estuaries, and at least in one case a severe infection with *Pdd* was apparently contracted by handling fish from a fresh water lake [16].

Soon after its first description as a pathogen in *C. punctipinnis*, isolates of the newly recognized species from human patients were also published [2]. One of the isolates originated from a patient whose case had been previously reported as tissue invasion by an unnamed marine Vibrio [17]. In the cases reported by Morris et al., the likely entry ports of *V. damsela* were injuries contaminated with sea water or getting in contact with infected marine animals [2]. In fact, all cases were wound infections, and no patient had a known underlying illness. All of them fully recovered, leading the authors to state that, “*V. damsela* does not, however, appear to have the virulence or pathogenic potential of *Vibrio vulnificus*, which has been associated with tissue necrosis, fasciitis, and bullae formation.” However, subsequent case reports implicated *V. damsela/Pdd* in severe and even fatal infections, e.g., [3,4,16,18]. In one early report, the case of an alcoholic 61 year old male with a history of pancreatitis and insulin-dependent adult onset diabetes has been described, who reported with a painful wound on his hand that had developed within hours after a slight injury while cleaning a catfish caught from a fresh-water lake [16]. The patient developed bullae over the painful, swollen and erythematous skin. Cultures and stains from tissue yielded Gram-negative, curved, rod-shaped bacteria, which were identified as *V. damsela*. While the affected tissue expanded and systemic effects of the infection became apparent including disseminated intravasal coagulation, renal failure and hypercalcemia, the patient deteriorated and died of a cardiac arrest nine days after the injury. Eight of eleven human cases of severe wound infections caused by *Pdd* that were compiled in a recent review of the literature were fatal [19]. Notably, three of them occurred in patients without any obvious underlying health problem. The average age of patients with fatal outcome was >65 ys and all of them were males. Whether this is due to a gender bias related to occupational or recreational activities or to physiological differences is unknown. Although there are few case reports in the literature reporting *Pdd* as the likely pathogen in humans, e.g., [18,19], the prevalence of such infections by *Pdd* is not known. Notably, in two cases there was a co-infection with *Vibrio harveyi* [5,19], another significant pathogen [20]. That *Pdd* may cause hyper-aggressive soft tissue infections and septicemia, even in otherwise healthy individuals [4,19], warrants investigation of the underlying patho-mechanisms. Previously, a phospholipase D/sphingomyelinase D, termed damselysin has been identified as a virulence factor [21,22,23], but only recently have PFT been discovered in *Pdd*.

## 2. Hemolysins and Hemolysin Genes in *Pdd*

### 2.1. Plasmid Encoded Hemolysins

Pathology and clinical course of necrotizing soft tissue infections (NSTI) caused by *Pdd* are reminiscent of conditions due to *Clostridium perfringens* [24,25,26], *V. vulnificus* [26,27] or *Streptococcus pyogenes* [26,28]. From early on, this suggested a major role of exotoxins in the pathogenesis of severe *Pdd*-infections. Because many secreted bacterial toxins, including toxins produced by the afore-mentioned species, exert hemolytic activity [24,25,29,30], a systematic analysis of 19 *V. damsela* strains had been performed which indeed led to the finding that 17 strains secreted a hemolytic factor active against murine red cells [23]. More importantly, high hemolytic activity correlated with low LD50 in subcutaneously (s.c.) infected mice (with LD50 ranging from 2 × 10^7^ to 9 × 10^8^ CFU), which usually died within 48 h after infection. The number of bacteria needed to kill 50% of the s.c. injected mice was more than two logs higher when culture supernatants of low hemolytic units were injected (hemolytic units ranged between 2 and 75.000). The study also showed that low Na^+^-concentrations in media promoted the release of hemolytic activity and marked differences in sensitivity to hemolysins between erythrocytes of various species was noted. Red cells from mice and rats turned out to be particularly sensitive, while human red cells were resistant [23]. Subcutaneous injections of the partially purified toxin, which had an app. MW of ~58 kDa recapitulated the gross pathology caused by infection with *V. damsela*. Physicochemical and biological characterization of the toxin [22], which was termed damselysin (Dly), showed that it is a sphingomyelinase D [21] and phospholipase D with activity against phosphatidylcholine and phosphatidylethanolamine [31]. Subsequently, the gene encoding damselysin was cloned for heterologous expression of this toxin in *Escherichia coli* [32]. Since then, little additional information on the function of this toxin has been published. In retrospect it appears that one of two hemolytic phenotypes (termed LZ, for large (hemolysis) zone) of *V. damsela*, associated with the first fatal case reported in the literature [16] may have expressed damselysin because in a CAMP (Christie, Atkins, and Munch–Petersen)-test [33] with *Staphylococcus aureus*, the LZ strain protected red cells from lysis by *S. aureus* β-toxin, a phospholipase C.

More than 20 years after *dly,* the gene encoding Dly, had been cloned [32], it was found that it is located on pPHHD1, a 153,429-bp conjugative plasmid isolated from highly hemolytic strain RM-71 [34]. The complete sequence of pPHDD1 was established [34], leading to the finding that it also encodes an orthologue of the small β-pore-forming *Vibrio cholerae* cytolysin (VCC); this gene was termed *hlyA*_pl_ (hemolysin A, encoded on a plasmid). A direct correlation between presence of pPHDD1 and hemolytic phenotype in a collection of *Pdd isolates* was observed. Hemolysis was markedly reduced in a mutant where both *dly and hlyA*_pl_ had been deleted, demonstrating the role of these pPHDD1-encoded genes in hemolysis. Although single mutants of either gene showed different levels of hemolysis reduction depending on the source of erythrocytes, hemolysis was not abolished in any of the single mutants, indicating that it results from additive effects of *hlyA* and *dly* [34]. Importantly, *dly* and *hlyA* proved to be necessary for full virulence in mice and fish. Therefore, these genes seem to play a role for virulence in homotherms and poikilotherms, and the acquisition of pPHDD1 has been considered a driving force during evolution of a highly hemolytic lineage within the subspecies.” [6]. A role of plasmids as carriers of virulence genes in other members of the family has been previously discussed [35].

Although the in silico translated sequence of *hlyA*_pl_ is 48% identical to VCC, significant differences exist. Most notably, the *hlyA*_pl_ gene of *Pdd* does not encode a β-prism domain, the C-terminal lectin domain in VCC, but just the β-trefoil domain (Figure 1). Accordingly, homology-based structure prediction returned a model closely resembling that of VCC but leaving unmatched VCC´s β-prism domain. In this regard, the structure of the *hlyA*_pl_-encoded protein is closer to *V. vulnificus* cytolysin [36,37]. A second distinguishing feature of the sequence pertains to the cytolysin domain, where at the narrow-point of the channel there is a serine in PhlyP (S341) but a bulkier tryptophane (W318) residue in VCC. Corresponding residues in the majority of orthologous cytolysins from other *Vibrio* spp. are of the VCC-type (W), although, apart from PhlyP, a few additional exceptions do exist, for instance cytolysin of *Vibrio fischeri.*

### 2.2. Chromosomally Encoded hlyA

Once the sequence of *hlyA*_pl_ and the genome of a *Pdd*-isolate were available, it became clear that a gene very similar to *hlyA*_pl_, is located on the chromosome. This gene, termed *hlyA*_ch_ appears to be present in all hemolytic strains [38]. Hemolysis was abolished in a *hly*A_ch_ deletion mutant of plasmid-less strain AR111 [38] and also in a triple mutant strain lacking functional *dly, hlyA*_pl_ and *hlyA*_ch_. Thus, dly, *hlyA*_pl_ and *hlyA*_ch_ are required for full hemolytic activity in strains carrying pPHDD1 [38]. All highly and medium hemolytic strains carry this virulence plasmid [39]. An isolate with amino acid substitution A575S in the β-trefoil region of *hlyA*_pl_ was of lower hemolytic activity, suggesting that such mutations in *hlyA*_pl_ and *hlyA*_ch_ might be responsible for medium hemolytic phenotypes in some pPHDD1-harbouring strains [39]. Strains containing only *hlyA*_ch_ were of low hemolytic power and non-hemolytic isolates lacked pPHDD1 and *hlyA*_ch_ altogether, or they contained a *hlyA*_ch_ pseudogene [39]. The chromosomal region encoding *hlyA*_ch_ shows sequence diversity suggesting that it is unstable [38].

## 3. Regulation of Hemolysin Production

### 3.1. Transcriptional Regulation 

By isolating a mini-Tn10 transposon mutant that showed a strong impairment in its hemolytic activity, a role of *rstB* (encoding a protein of a two component regulatory system) for the expression of *Pdd*´s major hemolysins was demonstrated [40]. The transposon disrupted a putative sensor histidine kinase gene (*rstB*), which together with *rstA* was predicted to encode a putative two-component regulatory system homologous to the *V. cholerae* CarSR/VprAB and *E. coli* RstAB systems. Reconstruction of the mutant by allelic exchange of *rstB* showed equal impairment in hemolysis, and complementation with a plasmid expressing *rstAB* restored hemolysis to wild-type levels [40]. Promoter expression analyses revealed that the reduced hemolysis in the *rstB* mutant was accompanied by a decrease in transcription of *dly, hlyA*_pl_, and *hlyA*_ch_ [40]. Thus, *rstB* encoded in the small chromosome regulates plasmid and chromosomal virulence genes. Further, it was found that reduced expression of the three major hemolysin genes correlated with a strong decrease in virulence in a sea bass model, demonstrating that RstB constitutes a master regulator of these hemolysins and plays a critical role in the pathogenicity of this bacterium. Results of this study represented the first evidence of a direct role of a RstAB-like system in the regulation of bacterial toxins, in keeping with its established role as a regulator of gene expression. Recently, the concept was confirmed in a work addressing the role of *rstA* [41]. An unexpected regulatory link between cytotoxin- and chemotaxis genes [42] will be discussed further below (paragraph 7.).

### 3.2. Secretion 

The major toxins of *Pdd* are released by the type II secretion system (T2SS), [43]. The T2SS, which is used by Gram-negative organisms only, uses a two-step process to sequentially transport cargo across the inner and the outer membrane. It employs a Sec or Tat-dependent first step to cross the inner membrane and ultimately secretes folded substrates [44]. A mini-Tn10 transposon mutant in a plasmid-less strain with impaired hemolysis was shown to contain an insertion in the *epsL* gene [43], which encodes a component of the T2SS required, for instance, for the release of cholera toxin by *V. cholerae*. [45]. Reconstitution of the *epsL* mutant in *Pdd* confirmed the role of *epsL* for secretion of the *hlyA*_ch_-encoded hemolysin [43]. Mutation of *epsL* in a pPHDD1-harboring strain lacking *hlyA*_ch_ caused almost complete loss of hemolytic activity against sheep erythrocytes, indicating that *epsL* is also important for secretion of the plasmid-encoded hemolysins [43]. Further, mutation of the putative prepilin peptidase gene *pilD* led to a marked loss of hemolysis, and reporter analyses suggested that impairment of hemolysin secretion in *epsL* and *pilD* mutants might affect hemolysin and T2SS gene expression at the promoter level [43]. Importantly, deletion of *epsL* or *pilD* caused a significant decrease in virulence for mice, which supports the conclusion that hemolysins secreted via T2SS serve as virulence factors in *Pdd* [43].

## 4. Diversity and Molecular Epidemiology of Hemolytic Phenotypes

In strains carrying pPHDD1, hemolysis appears to involve additive effects of *hlyA*_pl_ and *hlyA*_ch_, as well as synergistic effects of *dly* with *hlyA*_pl_ or *hlyA*_ch_ [38]. Dly-producing strains give a CAMP reaction, i.e., they produce synergistic hemolysis with strains secreting only the product of *hlyA*_pl_. 

Phylogenetic analyses suggested that gene duplication within *Pdd* following acquisition by horizontal transfer led to *hlyA*_pl_ and *hlyA*_ch_ [39]. A recent molecular epidemiology study of *Pdd* outbreak strains in marine rainbow trout farms supported the idea of extensive horizontal gene transfer and high degree of genetic diversity [46]. Outbreaks in fish farms may be caused by multiclonal populations of *Pdd* that co-exist in the environment; horizontal gene transfer might lead to intra-species diversity. The majority of isolates from fish outbreaks lack pPHDD1. The chromosome I-encoded *hlyA*_ch_ and two additional virulence genes encoding phospholipase (PlpV) and a collagenase (ColP) are critical for virulence and cytotoxicity of these pPHDD1-negative isolates in fish [47]. In contrast, *dly* and *hlyA*_pl_ appear to be the main virulence factors of *Pdd* in mice [38]. Therefore, recent efforts have focused on the characterization of the toxin encoded by *hlyA*_pl._


## 5. Phobalysin P

### 5.1. Stable Toxin-Complexes and Membrane Pores

Employing AR119, a *Pdd* strain with deleted *dly* and *hlyA*_ch_ genes [38], the toxin encoded by *hlyA*_pl_ was purified from extracellular products (ECP) by isoelectric focusing for initial characterization of biochemical and functional properties [36]. The protein has an isoelectric point of ~ 5.5 and in SDS-PAGE migrates as a single band, corresponding to an apparent MW of ca. 50 kDa [36]. Because the predicted MW of the pro-toxin is 68 kDa, this suggested that the species isolated from ECP represents the processed form of the toxin (i.e., devoid of the pro-domain). This assumption was confirmed by Edman degradation of the purified toxin, which yielded the N-terminal sequence N-term-valine-alanine-serine-aspartic acid-glutamine-C-term [36]. The purified protein is hemolytic and also cytotoxic for various nucleated cell types, including human keratinocytes [36] as well as mouse embryonal fibroblasts [48]. It is also toxic for AB.9 cells from *danio rerio*, cultured at 28 °C [36]. The protein encoded by *hlyA*_pl_ was termed PhlyP, for “**P**hotobacterial **ly**sin encoded on a **p**lasmid”. Mature PhlyP is expected to comprise 447 amino acids and to have a molecular mass of 50.47 kDa, in keeping with results from SDS-PAGE analysis of the purified protein. Recombinant pro-toxin (pPhlyP) expressed in *E. coli* recapitulates the cytotoxic effects of mature PhlyP [36], but toxicity is somewhat delayed in comparison to mature PhlyP, probably because proteolytic activation is required. Three lines of evidence support the notion that PhlyP forms trans-membrane pores as expected from a small β-PFT. First, on red cell ghosts it gives rise to complexes that resist degradation by trypsin and run as a band of ~250 kDa apparent MW in SDS-PAGE [36]; similar SDS-stable complexes are seen after treatment of ghosts with VCC [48]. Second, circular structures typical of membrane pore-complexes can be discerned on PhlyP-treated red cell membranes (“ghosts”) in transmission electron microscopy [36], again, similar to observations with other small β-PFT, e.g., *S. aureus* α-toxin [49]. Third, an effective diameter of pores created by PhlyP can be estimated at >1.2 and <3.0 nm based on osmo-protection experiments [36]. Hence, pore size of PhlyP is a little wider, in comparison to the VCC pore (<1.2 nm). Depletion of cholesterol from red cells by incubating them with methyl-β-cyclodextrin abolished hemolysis, which could be recovered by adding back cholesterol [36]. Similar observations have been previously made with related toxins [50].

### 5.2. Cytotoxicity

Treatment of human epithelial cells (HaCaT) with PhlyP causes a rapid drop of intracellular potassium levels and intracellular ATP concentrations [36,48]. Within twenty minutes, MAPK p38 and eIF2α become phosphorylated, closely followed by a collapse of intermediate filaments and attenuation of global translation [36], known downstream effects of these phosphorylation-events [51,52,53,54]. Furthermore, there is a breakdown and redistribution of actin-filaments, accompanied by rounding of cells and a marked mitochondrial shape change. These molecular signs of cellular stress recapitulate findings observed after treatment of cells with other small β-PFT, reviewed in [55,56,57,58], but some details observed with PhlyP are special and deserve mention: First is the speed at which potassium is lost from the cell. Whereas exposure to VCC causes a gradual efflux of potassium over 15 min, loss of potassium reaches its maximum within 2 min after treatment with PhlyP [48]. In line with estimated pore sizes in the low nm range, neither VCC nor PhlyP cause rapid egress of lactate dehydrogenase (LDH) from cells. However, influx of propidium iodide (PI) was readily observed with PhlyP, but not with VCC. Second, PhlyP provokes a steep increase of fluorescence in HaCaT cells loaded with the calcium-sensitive dye Fluo-8-AM, with an onset around 12 s after addition of toxin [48]. This increase was only moderately reduced in the presence of suramin a broad spectrum inhibitor of purine receptors, G-protein coupled receptors and other targets [48]. Although VCC also triggers a substantial increase of intracellular calcium concentration, this was much delayed (onset around 60 s) and virtually blocked by suramin [48]. Together, this suggested that PhlyP pores are of higher conductance for both potassium ions and calcium ions as compared to VCC pores, and that calcium influx in the case of VCC likely involves cellular channel forming proteins. That this difference can be at least partially attributed to channel width is suggested by the fact that replacing the amino acid at the presumed narrow point of PhlyP (serine) with a bulkier tryptophan residue, the corresponding amino acid in the VCC channel, greatly reduced rapid suramin-insensitive entry of calcium ions or PI [48]. Influx of calcium ions following attack by PFT has important implications for the response and fate of target cells [59], one important consequence of calcium influx being rapid membrane repair.

## 6. PhlyP as a Paradigm of Membrane Repair

### 6.1. Calcium Influx-Dependent Repair (CIDRE) of Cholesterol Dependent Cytolysins (CDC) Pores

In order to understand how studies on PhlyP contribute to the elucidation of membrane repair mechanisms, it is necessary to briefly consider current concepts: Early studies in sea urchin eggs uncovered the ability of nucleated cells to recover from mechanical wounding of their plasma membrane and showed that this depends on extracellular calcium [60,61]. Subsequent work showed that entry of calcium triggers the exocytosis of vesicles providing a conceptual frame work to explain membrane repair [62,63,64]. Meanwhile, it had been observed that cells can also recover from damage by pore forming toxins and mount productive responses, including activation of signaling mediators, transcriptional regulation and cytokine production [65,66,67,68]. Similar observations have been made with complement and perforin [69,70] and references therein. During the past decade mechanistic details of membrane repair after attack by PFT have been elucidated; for review see e.g., [71,72,73]. The majority of these studies have been performed using streptolysin O (SLO) or related CDC because the large non-selective pores permit, by using dye-exclusion from intact cells, the microscopic detection of membrane breaches. As observations are usually made at a very early time point after addition of toxin, it is tacitly assumed that cellular channel forming proteins or ruptures do not contribute to the influx of dye. At any rate, important insight into the consequences of membrane damage, e.g., [67,68], and various mechanism of CIDRE [72,73,74,75,76,77,78,79,80], has been gained by studies on CDC. In essence, these studies showed that removal of CDC pores from the plasma membrane is important for recovery of target cells and that this may involve uptake or shedding of membrane lesions. Another mechanism being discussed is clogging of lesions. Nonetheless, all of these studies agree on the central role of calcium influx for membrane repair, a fact which at times may lead scientists to falsely equate membrane repair and CIDRE.

### 6.2. Subversion of CIDRE 

This said, membrane repair mechanisms identified with CDC do not uniformly apply to other PFT. Thus, cellular recovery from damage by *S. aureus* α-toxin or aerolysin, small-β-PFT which have been classified on structural grounds as “hemolysin”-type and “aerolysin”-type toxins, respectively [81], is much slower than recovery following treatment with SLO or listeriolysin (LLO) [82,83]. It has been proposed that insufficient calcium-influx through α-toxin pores could be responsible [82]. Others have argued that this explanation is contradicted by data showing that small pore forming toxins, including α-toxin, may trigger massive influx of calcium into cells [59], but the reports that were referred to [84,85,86] leave open to what extent toxin-dependent calcium-entry actually occurred through toxin pores. This is important, as suggested by the fact that cells intoxicated with a pore former which does not trigger CIDRE are not rescued by simultaneous induction of calcium-influx with another agent [48,82], possibly reflecting the need for sufficient *local* rises of intracellular calcium. It is known that increases in cytosolic calcium concentrations after limited membrane damage are usually transient and confined in space [87]. Moreover, cellular channels may contribute to the effects of PFT, a field that deserves further attention [88,89,90,91].

### 6.3. Alternative Rescue Mechanisms 

As a matter of fact, repair of *S. aureus* α-toxin pores proceeds in the absence of extracellular calcium [92], violating the dogma that calcium-influx is key to membrane repair. Yet other mechanisms stand in, which in the case of *S. aureus* α-toxin have been characterized to some detail. Both α-toxin toxin monomers and oligomers are endocytosed, and the internalized toxin travels to the late endosomal compartment [93]. Clearance of oligomers from the plasma membrane is closely followed by metabolic recovery of target cells. Curiously, endocytosis of α-toxin involves translation-factor eIF2α which becomes phosphorylated by nutrient sensitive kinase GCN2 upon treatment of cells with α-toxin [52,94,95,96]. In epithelial cells, phospho-Ser51-eIF2α (p-eIF2α) marks sites of toxin-attack at the plasma membrane for uptake by a process that depends on PPP1R15B, a subunit of eIF2α-phosphatase that serves to target the catalytic subunit to the substrate, i.e., p-eIF2α [96]. PPP1R15B, which is constitutively located at the plasma membrane, is recruited to accumulations of p-eIF2α juxtaposed to membrane bound α-toxin. By virtue of its amphipathic α-helix, PPP1R15B impacts the curvature of the plasma membrane and may thereby lead to endocytosis of α-toxin [96]. Whereas internalization of SLO is dynamin-independent and seems to take the caveolar route [97]; internalization of α-toxin depends on dynamin [93] and may be related to macro-pinocytosis [98,99]. Again in contrast to SLO, α-toxin oligomers are not efficiently degraded by cells but released as exosome-like particles, toxosomes [93]. To sum, repair of *S. aureus* α-toxin pores differs in many aspects from repair of SLO-pores. Importantly, it is clearly independent of CIDRE. However, potassium efflux-triggered survival mechanisms, although at a slower rate, may ensure recovery of cells from attack by PFT that form small pores which do not trigger efficient CIDRE, e.g., [55,58,94,100,101,102,103,104,105]. 

### 6.4. PhlyP—A Small β-PFT that Triggers CIDRE

For many years, it could not be excluded that structural features common to small β-PFT, but not shared by CDC, might preclude rapid removal of membrane lesions and fast recovery [57]. The recent finding that PhlyP elicits CIDRE similar to CDC has ruled out this possibility [48]. Extracellular calcium, lysosomal exocytosis, acidic sphingomyelinase, increased endocytosis and caveolin, all appear to be involved in membrane repair after attack by PhlyP [48]. In contrast, phosphorylation of p38, and blebbing, although triggered by PhlyP, are dispensable in this case. That channel width is critical for sufficient entry of calcium ions through PhlyP pores and down-stream events, like lysosomal exocytosis, is strongly suggested by experiments with a single amino acid exchange mutant where serine 341 of PhlyP was replaced with tryptophan, the residue at the channel narrow point in VCC, a closely related PFT which, however, does not trigger CIDRE [48]. To sum, PhlyP, a bacterial small β-PFT, leads to rapid calcium influx and wounded membrane repair like responses. Therefore, not the affiliation with a structural family of PFT but sufficient influx of calcium through pores seems to be necessary for triggering rapid membrane repair. 

### 6.5. Technical Advantages of PhlyP for Studying CIDRE

As outlined above, PhlyP proved to be an insightful model to better understand function or failure of CIDRE. Moreover, from a technical point of view, PhlyP has a number of useful features (Figure 2): First, unlike CDC, which yield pores of different stoichiometry, PhlyP is expected to form a single species of pores (presumably 7mer, as inferred from VCC). This would simplify interpretation of data. Second, PhlyP pores do not permit egress of cytosolic proteins, which might impact membrane remodeling. This too simplifies interpretations. Along the same lines, LDH which does not escape through PhlyP pores, can be used as a marker to monitor secondary plasma membrane damage, which may interfere with observations of primary effects of PFT [91]. Last not least, the fact that cells perforated by PhlyP retain the calcium sensitive dye Fluo-8(-AM) facilitates dynamic monitoring of changes in calcium-concentrations in target cells, because analyses can be performed on entire cell populations and for extended periods of time. 

## 7. PhlyP Promotes the Association of *Pdd* with Epithelial Cells

### 7.1. Phlyp Enhances Adhesion

In short term co-culture experiments (15 min) under various conditions (different salinities and temperatures) *Pdd* was found to line free edges of HaCaT cells, human immortalized keratinocytes [36,42]. Deletion mutants deficient in phobalysin production showed significantly reduced adhesion. Other membrane damaging toxins have been implicated in bacterial adhesion [106,107,108,109,110]. In contrast to RTX (repeats in toxin)-protein-dependent adhesion [111], the mechanisms remain unclear in all these cases. The effect of PhlyP on the association of *Pdd* with epithelial cells is significantly reduced by dynasore [36], a selective inhibitor of the large 100 kDa GTPase dynamin [112,113]. EDTA a chelator of divalent cations and inhibitors of PI3K exerted similar effects [36]. Possibly, ongoing membrane repair increases the interaction of *Pdd* with cells. The mechanisms could be related to those implicated in the invasion of non-professional phagocytes by *Listeria monocytogenes* [114] and might involve manipulation of the actin cytoskeleton and macro-pinocytosis [98,99,114].

### 7.2. A Role of Chemotaxis Regulators

Under cell culture conditions, phobalysin-dependent adhesion did not require intact *cheA*, a conserved regulator of chemotaxis, but a potential role of chemotaxis emerged when experiments were done in media of low nutrient content [36,42]. Under these conditions, which more closely resemble the marine environment of *Pdd*, adherence to epithelial cells was markedly decreased by disruption of *cheA* [42]. One explanation could be *exotoxin-guided chemotactic orientation* (ECHO): Solute gradients emanating from cells damaged by PFT might serve as a trace for directional movement of *Pdd* towards its host. In support of this concept, efficient PhlyP-dependent adhesion involves co-operative action of cytotoxin- and chemotaxis-genes [42]. Only with intact *cheA* was adhesion of a hemolysin-deficient *Pdd*-strain significantly enhanced by exogenous PhlyP [42]. Interestingly, deletion of *cheA* also reduces the association of *L. monocytogenes* to Caco-2 cells [115]; whether LLO was involved is not known. Secretion of PhlyP is massive and fast [36], and so is the release of solutes from target cells [48]. Because even in the turbulent environment of oceans gradients of chemo-attractants released from decaying cells are sufficiently stable to be recognized by bacteria [116], ECHO is a reasonable scenario to explain *hlyA*_pl_- *and cheA*-dependent adhesion. The observation that disruption of *cheA* leads to reduced production of PhlyP could provide an alternative explanation. The machinery involved in chemotaxis and motility, or chemotactic motility itself, appears to impact hemolysin production by *Pdd* because treatment with phenamil (an inhibitor of the Na^2+^-channel of the flagellar motor) or disruption of *cheA* reduced the production of PhlyP and of PhlyP mRNA expression [42]. Because *cheA* belongs to a gene family encoding proteins that control gene expression [117], it is possible that it directly regulates transcription of *hlyA*_pl_. Alternatively, disruption of *cheA* or treatment with phenamil could interfere with some step further downstream in the process of PhlyP production. Besides temperature, salinity is the main a-biotic factor that determines abundance and distribution of marine *Vibrionaceae* [118,119,120,121]. This seems to be true for *Pdd* as well [122]. Swimming of wild type *Pdd* on soft agar and production of PhlyP and damselysin reach a maximum at low salinity but are almost nil at 3.5% NaCl, the average salt concentration of sea water [42]. A shift of high to low salinity seems to trigger a transition from some resting state to an active state, possibly reflecting a conversion of *Pdd* from a tranquil environmental modus to an aggressive infection modus in response to nutritional or environmental stimuli. A comprehensive identification of genes involved in host adaptation may reveal a more complex picture, as suggested by studies on *V. vulnificus* [122,123,124].

## 8. Perspective

### 8.1. Receptors

One area of future investigations on PhlyP will be the analysis of binding characteristics to target cell membranes (e.g., of human keratinocytes, or murine embryonal fibroblasts). The only information available today is the requirement for cholesterol in target cell membranes for binding and lytic activity toward red cells, which is reminiscent of VCC´s binding characteristics. Although experiments with liposomes indicate that there is no absolute requirement for a non-lipid receptor, it appears likely that such (a) receptor(s) exist for PhlyP.

### 8.2. Lectin Domain (LD)

The high resolution structure available for VCC [37] has provided a template for an approximation of PhlyPs structure. Sequence comparison and homology based modeling strongly suggest that PhlyP has a very similar structure as VCC [36] but in contrast to the latter features only a single lectin domain at its C-terminus. The functions for both LD in VCC have been delineated [125,126], which may help to analyze the role of PhlyP´s LD. Interestingly, truncation of the C-terminal LD in VCC eliminates lytic activity, but the single LD of PhlyP is sufficient to convey activity. 

### 8.3. Pro-Toxin Processing

Although mature functional PhlyP has been purified from ECP of *Pdd* cultures, PhlyP is presumably secreted as a pro-toxin. This raises the question when and where the precursor is cleaved, and what enzymes are involved. The comprehensive analysis of secretomes [127] has become feasible and may lead to the identification of proteases activating PhlyP. A subset of proteins secreted by the *Pdd* has been recently characterized, including several proteases [41], but their role in activation of pPhlyP remains to be studied. Like VCC, PhlyP is expressed as a pro-toxin, but the cleavage site of VCC protoxin (pVCC) is not conserved in pPhlyP [36]. It might be for this reason that inhibitor of tumor necrosis factor alpha processing-2 (TAPI-2), which prevents processing of pVCC by cellular proteases [128], could not block the hemolytic activity of pPhlyP. The cleavage site in pPhlyP seems to be located at position 155 of the amino acid sequence, as determined by Edman degradation of a gel band that corresponded to the hemolytic active (mature) form of PhlyP [36]. 

### 8.4. Interaction of PhlyP and Other Factors

The amino acid sequence of PhlyP is 95% identical with the sequence of PhlyC, the gene product of *hlyA*_ch_. ECP of *Pdd* strains devoid of PhlyP and Dly still exhibit significant cytotoxicity, whereas a triple mutant devoid of PhlyP, PhlyC and Dly does not. This suggests that PhlyC is a cytotoxin, functionally similar to PhlyP [38]. Unlike PhlyP, however, PhlyC has not been characterized, and functional comparisons remain to be done. One question in that context is whether PhlyP and PhlyC can form hetero-oligomers, and whether these are endowed with new properties. Because the promoters of these two toxins are different, their transcription could be regulated by different cues. The characterization of phobalysins will be prerequisite to a better understanding of potential synergy between these toxins and other virulence factors.

### 8.5. Engineering Phobalysins

PFT have potential as tools or drugs. [129,130,131,132]. Several domains/properties of phobalysins could be manipulated to tailor a recombinant toxin to specific needs: The recognition site could be modified to alter proteolytic activation, LD for changing binding characteristics and residues lining the channel for modulating conductance and selectivity.

## 9. Conclusions

To sum, PhlyP is a plasmid-encoded small β-pore forming toxin of *Pdd*, a pathogen of vertebrate and invertebrate hosts. PhlyP has been shown to serve as a virulence factor in mice and fish. It represents the first paradigm of a bacterial small β-pore forming toxin which can elicit calcium-influx dependent membrane repair. Further, at serum level salinity PhlyP promotes the association of *Pdd* with epithelial cells through both chemotaxis-dependent and -independent effects.

Small β-PFT orthologous to PhlyP are expressed by many *Vibrionaceae*, which cause a range of quite different diseases [1,27,133,134], suggesting that these cytolysins do not explain the specific pathology caused by cognate bacteria. This prompts the question for the principal function of these rather conserved PFT. An emerging theme from studies on PhlyP and other toxins is that they might serve to promote the interaction of bacteria with their target cells, which might involve remodeling of plasma membrane and cytoskeleton during membrane repair. In addition, a chemotaxis-dependent *attack-and track* mechanism may play a role in motile bacteria producing membrane-damaging toxins: Membrane attack by PhlyP would lead to the leakage of chemo-attractants from cells; and bacteria would track the concentration gradients towards target cells (Figure 3). *Pdd* may serve as a model to study how such bacteria integrate environmental factors, chemotaxis and host-responses to their PFT to efficiently access a host. In addition to delineating mechanistic details, an important goal will be to verify emerging concepts in vivo. 

## Figures and Tables

**Figure 1 toxins-11-00412-f001:**
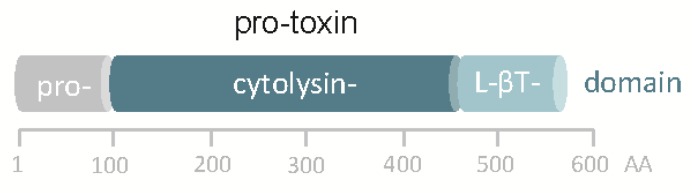
Domain structure of pPhlyP. Scale below scheme indicates amino acid positions. L-βT denotes lectin β trefoil.

**Figure 2 toxins-11-00412-f002:**
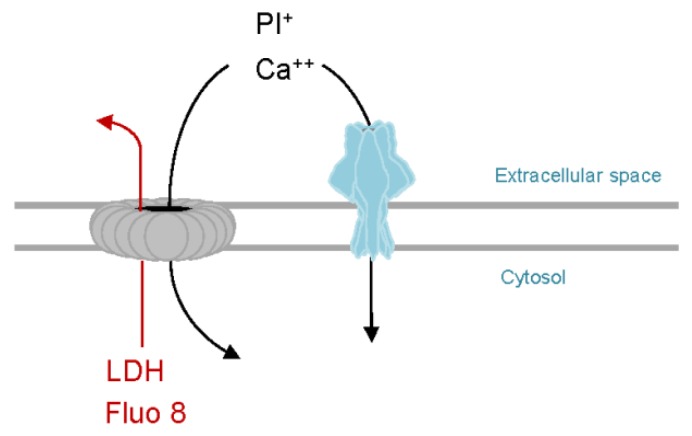
PhlyP as a tool to study membrane repair. Like SLO or other CDC (grey), PhlyP pores (light blue) permit entry of propidium iodide and calcium ions into target cells. Therefore, membrane damage can be monitored using PI, and calcium influx dependent repair mechanisms (CIDRE) are triggered in both cases. In contrast to CDC, PhlyP does not cause efflux of the calcium sensitive dye Fluo-8 or of the cytosolic marker LDH. Retention of Fluo-8 by PhlyP-treated cells can be exploited to measure changes of intracellular calcium ion concentrations in entire cell populations and over extended periods of time. Efflux of LDH can be used to detect secondary damage.

**Figure 3 toxins-11-00412-f003:**
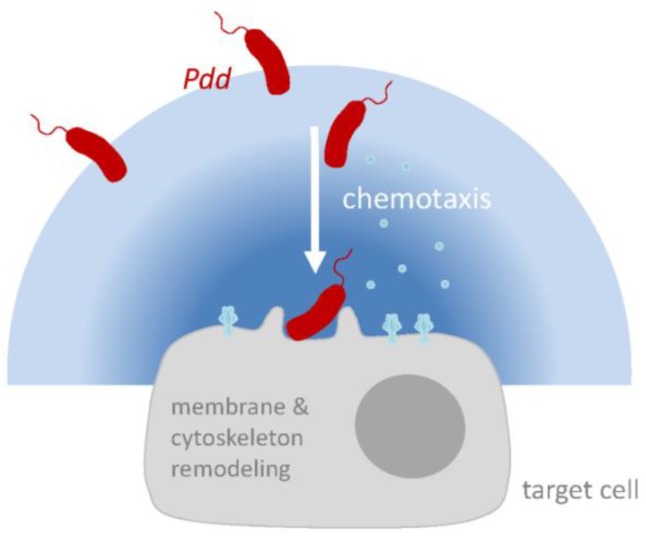
Hypothetical model of PhlyP-dependent association of *Pdd* with epithelial cells. Membrane perforation by PhlyP triggers signaling pathways in epithelial cells which enhances the association of *Pdd* to cells; this may involve remodeling of cytoskeleton and plasma membrane in target cells. In addition, at low salinity, cytotoxins and chemotaxis apparatus cooperate to enhance the association of *Pdd* with target cells. Release of chemo-attractants following membrane damage by PhlyP may thereby be involved.

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
