# Peer review of "Phobalysin: Fisheye View of Membrane Perforation, Repair, Chemotaxis and Adhesion"

_toxins, 2019, doi:10.3390/toxins11070412_

Round 1

Reviewer 1 Report

File has been attached.

Author Response

The helpful comments of the Reviewer were appreciated. Below find our point by point response (in italics), with lines where changes have been applied (revised manuscript) indicated and underscored.

Toxins_526721_report

Point by point response

Page: 1 - Abstract – Life threatening infection in humans?

These infections (soft tissue infections and septicemia) have been specified in the revised manuscript.           line 8

- Insightful model?

PhlyP is the first small b-PFT shown to trigger calcium influx dependent repair, demonstrating that CIDRE is not confined to the large pore forming cholesterol dependent cytolysins. This result and data from mutational analysis as well as comparison with VCC (see paragraph 6.4 and reference 48) make it an insightful - or revealing - model for function and failure of calcium-dependent repair. We have replaced the word“insightful” with “revealing” to enhance clarity.

line 12

- ‘Secretome’ – ‘it defines proteins secreted into extracellular space’ (Agrawal, GK, Proteomics, 2010, 10(4), 799-827. - My query is whether all the secreted proteins were characterized?

Pdd proteins secreted into the extracellular space have been analysed (Terceti et al. ref.41); and the authors used the term secretome in that context. However, because this study did not analyse “the global group of secreted proteins…” (Agrawal et al. Proteomics, 2010, 10(4) 799ff.), we have modified the text of our abstract (line 20).

Further, as a reference for definition and methodology of comprehensive secretome analyses we quote the above mentioned work by Agrawal et al. (Ref. 127) lines 422-423,

Major criticisms –

Which infections caused by this bacterium are life threating in humans?

soft tissue infections and septicemia (line 8)

This is a review article, and so the reader expects to get a brief description of the most important and relevant conclusions. This is missing.

In the revised version of the manuscript, the most important and relevant conclusions are summarized in the first paragraph of the Conclusions section (lines 447-452).

Pl. note that important conclusions have also been highlighted in the Abstract and the Key contribution section.

Line 173 – 184 – (Major criticisms) In this section, different results have been mentioned, however the authors give no citations of any of these.

Citations for the different results have been inserted (lines 185, 186, 187, 189, 192, 194).

Missing reference(s) (especially from line 205 -224) in this section – again, a major criticism.

Missing references are given in lines 211, 215, 218, 220, 221, 225, 229, 231, 232 and 233.

In this article there is no indication about the cell specificity of PhlyP’s, but the authors proposed, on line 390, that ‘future investigation of PhlyP … binding characteristics to target cell membranes’.

PhlyP exerts activity against various cell types (red cells, human keratinocytes, murine fibroblasts as well as AB cells from fish), see lines 219, 220 and 406.

Reference missing or incomplete (line 35, 50, 62, 101, 120, 145, 150, 361, 366, 369, 414,

Missing references are given in lines 38,53,66,106,128,151,159,375,380,383/384,435.

Comments, questions and minor criticisms –

Line 33 and 35 – ‘variety of disease’ and ‘serious infections in humans’ – What are these diseases and/or serious infections? These include gastrointestinal disease, skin ulceration, soft tissue infections, septicemia. lines 35 and 36.

Line 49 – ‘10E7-10E8 bacteria…’ – What does this mean? Is it the strain specification?

10E7-10E8 (E for exponent) denotes the number of bacteria used. We have replaced this notation with 107-108.      line 51

Line 66-67 – ‘However subsequent case reports…’ – given the brief details of the case, this is very unclear.

In the revised version of the manuscript we have added information on that case, specifically, results from stains and culture, and the identification of the isolate as V. damsela

lines 75-76

Line 81 -83 – ‘The fact that Pdd may cause hyper-aggressive …’ – Any proof or reported evidence of this statement?      See references 4 and 19, line 87

Which phospholipase is Damselysin? Damselysin is a phospholipase D/sphingomyelinase D.

line 88

Line 95 – What was the LD50 for s.c. (acronym for subcutaneous not specified before) mice The range of LD50 is given (line 100), and “subcutaneously” has been spelled out before use of the acronym (line 100).

Line 110 – What is CAMP-test? This needs clarification before the use of abbreviation.

CAMP has been spelled out and the reference has been given (lines 115-117).

Line 112 – What is dly and its function? dly is the gene encoding damselysin (Dly).

 line 118

Line 117 - hlyApl – Again, full details required before using abbreviations. …..This gene was termed hlyApl (hemolysin A encoded on a plasmid).            lines 121-122

What kind of mutant are these? It is not at all clear. The mutants are made by deletion of the genes.      line 124

Line 144 – Strain description is missing. Description of strain and reference are given

lines 150-151

Line 146 – 150 – Which substitutions resulted in the change in haemolytic activities?

The information is provided in lines 153-156 of the revised manuscript

Which strain contains only hlyAch and lacks pPHDD1 etc. that results in variations in their activity? Strain AR111 contains only hlyAch and lacks pPHDD1.      Line 151

Overall this section is very poorly presented. The text of this section (section 2.2; lines 147-159) has been amended and missing references and lacking informations are given.

Line 202 – The strain information is missing. Details of that strain and the reference is provided. line 211

Line 261 -262- ‘--- mount productive responses’ – A brief clarification of this response process is needed. Done.             lines 273-274

Line 275 – What is CIDRE equation in membrane repair? Our statement that is alluded to was “Yet, all of these studies agree on the central role of calcium influx for membrane repair, a fact which at times led to equation of CIDRE and membrane repair”. This sentence points to the fact that the term ”membrane repair” is sometimes used as if it has the same meaning as “calcium influx dependent repair” (thus ignoring the fact that membrane repair can be CIDRE-independent). The text has been modified to make this clear    line 287

Line 300 – ‘PPP1R15B, the target subunit eIF2-phosphatase’ – A brief description of this subunit is necessary for the readers. Done.       line 313

Line 328 – 330 – Is there any experimental evidence to support the proposition that the PhlyP pore would be a heptamer, based on VCC? There are examples of other small -PFT with different stoichiometry e.g. hexamer, nonamer etc..

Right. To our knowledge there is no direct experimental evidence for this proposition (hence, the phrasing “as inferred from VCC”). We have toned down the next sentence to now read “This would simplify interpretation of data”                   line 343

My next issue is that the authors have claimed, without any experimental evidence, that ‘PhlyP has been proved to be an insightful model to better understand function or failure of CIDRE’ (line 327 – 328). Experimental evidence supporting the contention that “PhlyP is an insightful model to better understand function or failure of CIDRE ….” is provided in reference 48, and is discussed in paragraph 6.4. lines 325-338. (In essence, the paper by von Hoven et al. shows that PhlyP triggers CIDRE whereas the closely related VCC does not. Further, analyses using single amino acid exchange mutants indicate that channel width is critical for both toxicity and efficient function or failure of CIDRE)        See also (response to) point 2.

Line 348 – The details of co-culture experiments are missing. Details are provided in the revised manuscript   lines 361-363

Line 350 – What is RTX? RTX stands for “Repeats in Toxin”         line 365

Line 369 – ‘target cells’ – Which cells? human keratinocytes or murine fibroblasts ; line 406

Line 412 What is PhlyC, and how similar is it to PhlyP ? The acronym PhlyC is explained and the degree of identity at the amino acid level is given            lines 432-433

The entire manuscript has been spell checked.

Reviewer 2 Report

This review manuscript is focused on presenting multiple biological, biochemical, and biophysical features of the photobacterial lysin encoded on a plasmid (Phobalysin P – PhlyP), a small beta-pore forming toxin from Photobacterium damselae subsp. damselae (Pdd). A major interest in this work stems in the fact that this toxin is an emerging pathogen of fish and other marine animals with a significant potential for causing severe infections in humans. Moreover, the toxin possesses a large variety of unique, intricate biochemical and biophysical characteristics; their detailed description warrants an enhanced appeal of the presented work to a broad scientific audience.

The review includes a systematic analysis of the diverse hemolysins and hemolysin genes in Pdd together with regulation and secretion, followed by a focus on PhlyP. In this line, they present experimental evidence in support of a mechanism that comprises pore formation and assess the ionic and molecular transport through PhlyP - permeabilized membranes by comparative analyses with other pore-forming toxins. Next, details on the Ca2+-influx dependent repair mechanism within the context of membrane repair pathways employed by cells under exposure to other toxins is provided. In addition, the manuscript presents evidence suggesting that the toxin itself contributes to the association between Pdd and target cells by exotoxin-guided chemotactic orientation.

The manuscript is well structured and the scientific information is gradually presented from a historical perspective. The references are well-chosen and span more than two decades of investigations relevant for this topic. The review has a good readability and only minor typos (i.e., missing spaces and full stops before and after the references) have been observed. Therefore, I consider that the manuscript is appropriate for publication in the journal Toxins after minor text edits (typos).

Author Response

We appreciate the positive evaluation of the manuscript by this reviewer.

Demanded corrections (missing spaces and full stops before and after the references) have been made.

Reviewer 3 Report

This review provides a good introduction of the field and provides a great summary of the Phobalysin: gene, gene regulation. function and possible working model.  One minor suggestion: Give attack-and track mechanism a brief definition, a sentence or two.

Author Response

We appreciate the very positive evaluation of the manuscript by the Reviewer.

-“One minor suggestion: Give attack-and track mechanism a brief definition, a sentence or two.”

Done: “a chemotaxis-dependent attack-and track mechanism may play a role in motile bacteria producing membrane-damaging toxins: membrane attack by PhlyP would lead to the leakage of chemo-attractants from cells; and bacteria would track the concentration gradients towards target cells (Figure 3).”           lines: 458-462

Round 2

Reviewer 1 Report

This version looks much better than first one.